# *Clostridium perfringens* Associated with Foodborne Infections of Animal Origins: Insights into Prevalence, Antimicrobial Resistance, Toxin Genes Profiles, and Toxinotypes

**DOI:** 10.3390/biology11040551

**Published:** 2022-04-01

**Authors:** Mahmoud M. Bendary, Marwa I. Abd El-Hamid, Reham M. El-Tarabili, Ahmed A. Hefny, Reem M. Algendy, Nahla A. Elzohairy, Mohammed M. Ghoneim, Mohammad M. Al-Sanea, Mohammed H. Nahari, Walaa H. Moustafa

**Affiliations:** 1Department of Microbiology and Immunology, Faculty of Pharmacy, Port Said University, Port Said 42511, Egypt; micro_bendary@yahoo.com or; 2Department of Microbiology, Faculty of Veterinary Medicine, Zagazig University, Zagazig 44511, Egypt; mero_micro2006@yahoo.com or; 3Department of Bacteriology, Immunology and Mycology, Faculty of Veterinary Medicine, Suez Canal University, Ismailia 41522, Egypt; rehameltrabely@gmail.com or; 4Veterinary Hospital, Faculty of Veterinary Medicine, Zagazig University, Zagazig 44511, Egypt; ahmed_vet8_2007@yahoo.com; 5Milk Hygiene Food Control Department, Faculty of Veterinary Medicine, Zagazig University, Zagazig 44511, Egypt; reemalgendy90@gmail.com; 6Air Force Specialized Hospital, Cairo 19448, Egypt; nahla.elzohairy@pharm.mti.edu.eg or; 7Department of Pharmacy Practice, College of Pharmacy, AlMaarefa University, Ad Diriyah 13713, Saudi Arabia; mghoneim@mcst.edu.sa; 8Pharmaceutical Chemistry Department, College of Pharmacy, Jouf University, Sakaka 72341, Saudi Arabia; 9Department of Clinical Laboratory Sciences, Najran University, Najran 66216, Saudi Arabia; mhnahari@nu.edu.sa; 10Microbiology and Immunology Department, Faculty of Pharmacy, Helwan University, Cairo 19448, Egypt; walaa_hassan@pharm.helwan.edu.eg

**Keywords:** *C. perfringens*, MDR, antimicrobial resistance genes, toxin gene profiles, toxinotypes

## Abstract

**Simple Summary:**

Recently, a crisis derived from foodborne infections, especially those are associated with food from animal origins caused by *Clostridium perfringens* (*C. perfringens*), has worsened. Unfortunately, the solutions to this crisis were restricted by an evolved resistance to antimicrobial agents. Therefore, we try to warn the world population about the hazards associated with this pathogen. The high diversity and polyclonality of *C. perfringens* strains depicted in our study show the urgent need to advance programs to control *C. perfringens* associated with foodborne infections. Additionally, the findings presented in this study are also of clinical importance, assisting in understanding the prevalence, origin, reservoir, and evolution of antimicrobial resistance of *C. perfringens* for establishing the control of this pathogen.

**Abstract:**

Several food-poisoning outbreaks have been attributed to *Clostridium perfringens* (*C. perfringens*) worldwide. Despite that, this crisis was discussed in a few studies, and additional studies are urgently needed in this field. Therefore, we sought to highlight the prevalence, antimicrobial resistance, toxin profiles, and toxinotypes of *C. perfringens* isolates. In this study, 50 *C. perfringens* isolates obtained from 450 different animal origin samples (beef, chicken meat, and raw milk) were identified by phenotypic and genotypic methods. The antimicrobial susceptibility results were surprising, as most of the isolates (74%) showed multidrug-resistant (MDR) patterns. The phenotypic resistance to tetracycline, lincomycin, enrofloxacin, cefoxitin/ampicillin, and erythromycin was confirmed by the PCR detections of *tet*, *lnu*, *qnr*, *bla*, and *erm*(B) genes, respectively. In contrast to the toxinotypes C and E, toxinotype A prevailed (54%) among our isolates. Additionally, we found that the genes for *C. perfringens* enterotoxin (*cpe*) and *C. perfringens* beta2 toxin (*cpb*2) were distributed among the tested isolates with high prevalence rates (70 and 64%, respectively). Our findings confirmed that the *C. perfringens* foodborne crisis has been worsened by the evolution of MDR strains, which became the prominent phenotypes. Furthermore, we were not able to obtain a fixed association between the toxinotypes and antimicrobial resistance patterns.

## 1. Introduction

Several outbreaks have been attributed to foodborne infections, which cause several life-threatening diseases and public health problems. This crisis has grown and become more complex due to the emergence of multidrug-resistant (MDR) fungi [1] and bacteria [2] (i.e., *Salmonella* species, *Staphylococcus aureus*, *Campylobacter jejuni*, and *Listeria monocytogenes*) [3,4,5,6]. *Clostridium perfringens* (*C. perfringens*) is among the most important foodborne pathogens worldwide [7]. Recently, *C. perfringens* foodborne outbreaks were diagnosed during the Panhellenic Handball Championship for children [8]. The *C*. *perfringens* is a Gram-positive anaerobic bacterium that is able to form spores under unsuitable conditions, and it has spread widely in the environment [9]. Additionally, it occurs within the normal animal gut flora and becomes pathogenic upon a disturbance in the balance of the gut microbiota [10]. Moreover, stress conditions, starvation, and the continuous administration of antibiotics or anthelmintic drugs may increase the pathogenic power of this pathogen [11]. 

Approximately 13% of the gastrointestinal foodborne outbreaks have been associated with *C. perfringens* infections [12]. *C. perfringens* foodborne infections are always associated with meat and poultry products. The meat products can be contaminated with this pathogen during slaughtering via the contaminated surface or the contact of carcasses with feces [13,14]. The heat resistance of *C. perfringens* is associated with the formation of spores that can germinate at temperatures ranging from 15 to 55 °C [15]. The standard food service practices should be followed up in order to prevent the spread of this pathogen [16]. Therefore, it is recommended to cook food until the internal temperature reaches 70 °C. Notably, the gastrointestinal infection with *C. perfringens* in animals and humans occurs due to the production of potent exotoxins [10]. Accordingly, *C. perfringens* can be serotyped into five groups (A to E) based on the production of specific exotoxins [alpha (α), beta (β), epsilon (ε), and ι (iota)] [17]. The α-toxin, encoded by a plasmid mediated *cpa* gene, is associated with serotype A, as well as all other serotypes of *C. perfringens* [18]. Meanwhile, the serotype B harbors *cpb* and *etx* plasmid-mediated genes that encode β- and ε-toxins, respectively. The β- and ε-toxins are associated with serotypes C and D, respectively [19], but serotype E has the plasmid-mediated *iap* gene, which produces ι-toxin [20]. All serotypes can contain *cpe* and *cpb2* genes, which produce *C. perfringens* enterotoxin (CPE) and *C. perfringens* beta2 toxin (CPB2) [21]. 

There are two general antimicrobial resistance mechanisms of *C. perfringens,* including the mutation of inherent genes or acquisition of resistance gene(s) [22]. The potential increase in the antimicrobial resistance of *C. perfringens* has been raised recently, with several reports announcing that most *C. perfringens* were MDR strains [23,24]. The resistance to tetracycline through TetA(P) protein, which regulates tetracycline active efflux, was common [25]. In addition, over 50% of *C. perfringens* isolates were resistant to lincomycin. However, 25% of this resistance was attributed to the expression of the *lnu* gene [26]. It is noteworthy that higher minimum inhibitory concentration (MIC) values of amoxicillin and ciprofloxacin were recorded due to the presence of the β-lactamase (*bla*) and quinolone (*qnr*) resistance genes, respectively [27]. In the same context, the macrolide-resistant *C. perfringens* may act as reservoirs for the *erm* gene, which assists in its conjugal transfer [28]. Owing to the increase in the emerging threat of foodborne-associated *C. perfringens* infections, we explored the prevalence of *C. perfringens* in food chains and spotlighted the evolution hazards of this pathogen in addition to the wide spread of MDR/toxigenic phenotypes in order to alarm the health organizations, especially in Egypt, to perform their duty and fulfill their responsibility.

## 2. Materials and Methods

### 2.1. Sample Collection 

A total of 450 raw milk, beef, and chicken meat samples (150 each) were collected from various supermarkets in two different Governorates in Egypt. Thirty raw milk, beef, and chicken meat samples (10 each) were obtained from 8 and 7 different supermarkets in Sharkia and Port Said Governorates, respectively (Appendix A). 

### 2.2. Isolation, Enumeration and Phenotypic Identification of C. perfringens

The collected samples were prepared according to the procedure recommended by the International Commission on Microbiological Specifications for Foods [29] to achieve ten-fold serial dilutions. Twenty-five grams or millimeters of each meat or milk sample were suspended into 225 mL of sterile peptone water (Oxoid, Basingstoke, UK)). The sample homogenates underwent heat shocking at 80 °C, in a water bath, for 15 min, to kill the non-spore forming aerobic bacteria, and then 1 mL of each sample was inoculated into 9 mL of fluid thioglycolate broth medium (Oxoid, UK), from which further ten-fold dilutions were prepared. A total of 1 mL from each of the previously prepared dilutions was streaked onto tryptose sulfite cycloserine (TSC) agar (Oxoid, Basingstoke, UK) plates, and all plates were incubated at 37 °C for 24 h in an anaerobic jar HP-11 (Oxoid, Basingstoke, UK), with gas generating kits (Oxoid, Basingstoke, UK). The plates were observed for the growth of *C. perfringens*, which was evident by the presence of characteristic black colonies [30]. The plates showing black colonies were selected, then the colonies were counted, and the results were interpreted as colony-forming units (CFU) per gram or mL of the sample. The presumed colonies on the agar plates were further subjected to purification by sub-culturing on TSC agar plates under the previously described conditions. The purified isolates were then identified by considering their cultural and morphological features, motility testing, double hemolysis on blood agar, and reverse Christie–Atkins–Munch–Petersen (CAMP) test. Moreover, the isolates were confirmed on the basis of some biochemical tests such as catalase, oxidase, nitrate reduction, indole, lactose fermentation, gelatin liquefaction, iron milk, and lecithinase production following the instructions specified in Bergey’s manual [31]. Additionally, further confirmation of the preliminary identified *C. perfringens* isolates was carried out using the commercially available API 20 A test system (bioMérieux, Marcy-l’Etoile, France) for identification of anaerobic bacteria according to the manufacturer’s instructions.

### 2.3. Genotypic Characterization of C. perfringens Isolates 

The DNA of *C. perfringens* isolates was extracted using a QIAamp DNA Mini Kit Qiagen GmbH, Hilden, Germany) following the manufacturer’s instructions. The molecular characterization was conducted depending on the amplification of a unique region of *C. perfringens 16S rRNA* gene using species-specific PCR primers, ClPER-F (5′-AGATGGCATCATCATTCAAC-3′) and ClPER-R (5′-GCAAGGGATGTCAAGTGT-3′) [32]. The amplification of the target sequence was performed according to the following protocol: one cycle for 2 min at 94 °C, followed by 35 cycles of denaturation at 94 °C for 30 s, annealing at 56 °C for 30 s and extension at 72 °C for 45 s, and finally one cycle at 72 °C for 2 min. The *C. perfringens* ATCC 3626 and *E. coli* ATCC 25922 strains were used as positive and negative controls, respectively, during all PCR runs.

### 2.4. Antimicrobial Susceptibility Testing

The *in vitro* antimicrobial susceptibility patterns of all confirmed *C. perfringens* isolates were tested against ampicillin, amoxicillin/clavulanic acid, cefoxitin, enrofloxacin, imipenem, chloramphenicol, lincomycin, metronidazole, erythromycin, and tetracycline, using a broth microdilution method to determine the MIC values (Appendix A). Double-fold serial dilutions (0.125–512 μg/mL) of the tested antimicrobials were prepared in a sterile microtiter plate, and a fresh *C. perfringens* culture adjusted to 5 × 10^5^ CFU/mL was added to each dilution. The plates were then incubated at 37 °C for 48 h under anaerobic conditions. Sterile broth and *C. perfringens* ATCC 3626 cultures were included as negative and positive controls in each run, respectively. The MIC values were determined according to the Clinical and Laboratory Standards Institute (CLSI) [33,34]. The MDR was defined as the resistance to at least one agent in three or more classes of the investigated antimicrobials. 

### 2.5. Typing of C. perfringens Toxins 

*Clostridium perfringens* toxins were typed using dermonecrotic tests in albino guinea pigs [35]. On the right side of guinea pig, a 0.2 mL of trypsinized 48 h supernatant of each *C. perfringens* culture was intradermally injected and the neutralized culture was injected on the left side in the same manner. The results were interpreted according to the degree of dermonecrotic reaction and its neutralization [36]. The neutralization tests were then performed in each albino guinea pig [37], using diagnostic *C. perfringens* antitoxin types A, B, C, D, and E (Burroguns, Welcome, Beckenham, London, UK).

### 2.6. Molecular Detection of C. perfringens Toxin and Antimicrobial Resistance Genes 

The extracted DNAs of MDR and toxigenic *C. perfringens* isolates were subsequently subjected to PCR approaches to detect their relevant antibiotic resistance and toxin genes. Uniplex PCR assays were carried out to detect *tet(K)*, *tet(L)*, *tet(M)*, *lnu(A)*, *lnu(B)*, *erm(B)*, *bla*, *qnrA*, and *qnrB* genes associated with tetracycline, lincomycin, erythromycin, β-lactams, and enrofloxacin resistances, respectively as previously detailed [28,38,39,40,41,42]. Moreover, toxinogenic genotyping of *C. perfringens* was performed using a multiplex PCR procedure to detect *C. perfringens* alpha (*cpa*), beta (*cpb*), epsilon (*etx*), iota (*iA*), and enterotoxin (*cpe*) genes [43]. A uniplex PCR protocol was used for the amplification of the beta2 toxin (*cpb2*) gene [44]. The PCR reactions occurred in a total volume of 25 μL consisting of 12.5 μL ofDreamTaq TM Green Master Mix (2X) (Fermentas, Inc. Hanover, MD, USA), 1 μL of each primer (20 pmoL (Sigma-Aldrich, Co., St. Louis, MO, USA)), 5 uL of DNA template, and 5.5 μL of PCR-grade water. The primer sequences, product sizes and annealing temperatures used to amplify target resistance and toxin genes of *C. perfringens* are shown in Table 1. Ten microliters of amplified PCR products were subjected to gel electrophoresis in 1.5% agarose gel and visualized after staining with ethidium bromide (Sigma-Aldrich, Co., St. Louis, MO, USA), under UV illumination. Our PCR results were validated using both positive and negative controls. The DNAs from *C. perfringens* isolates, which harbor the tested genes, were used as the positive controls. Sterile saline was used as a negative control. The positive control DNAs were provided by the National Laboratory for Veterinary Quality Control on Poultry Production (NLQP).

### 2.7. Statistical Analyses 

The results of antimicrobial susceptibility test, toxins genes’ profiles, and toxinotypes distribution were plotted in a heatmap through the GraphPad software (version 8.0.1, GraphPad software Inc., LA Jolla, CA, USA). The other data were analyzed using the following R packages: *heatmaply*, *corrplot*, *ggpubr*, and *hmisc*.

## 3. Results

### 3.1. Prevalence, Loads, and Characterization of C. perfringens Isolates 

Out of the 450 samples tested in this study, 50 were contaminated with *C. perfringens* with an overall prevalence rate of 11.1%. The samples from Sharkia governorate were more infected with *C. perfringens* (14.2%, 34 out of 240) than those from Port Said governorate (7.6%, 16 out of 210). The highest *C. perfringens* prevalence rate was found in chicken meat (12.6%, 19 out of 150), followed by raw milk (10.6%, 16 out of 150) and beef (10%, 15 out of 150). Interestingly, the number of *C. perfringens* in positive samples exceeded 10^2^ CFU/g or /mL with mean values of 1.2 × 10^4^, 2.4 × 10^3^ and 9.7 × 10^2^ CFU/g or /mL in the examined chicken meat, raw milk, and beef samples, respectively. The isolates were presumptively identified as *C. perfringens* on the basis of their cultural characters (Appendix A) and morphological and biochemical features. Moreover, the API 20 A test and genetic detection of the specific *16S rRNA* gene confirmed the identity of all recovered *C. perfringens* isolates.

### 3.2. Antimicrobial Susceptibility Results 

The highest resistance of *C. perfringens* isolates was recorded for tetracycline (84%) and erythromycin (72%). In turn, most of our isolates (70%) were sensitive to metronidazole and amoxicillin/clavulinic acid (Figure 1 and Figure 2). Tetracycline was the least effective drug for *C. perfringens* isolates, regardless of the source (beef, chicken meat, or milk) with resistance percentages of 80, 84.2, and 87.5%, respectively. Meanwhile, amoxicillin/clavulinic acid was the most effective drug for *C. perfringens* isolates in beef and chicken meat with susceptibility percentages of 73.3 and 68.4%, respectively. The highest susceptibility patterns for *C. perfringens* milk isolates were observed for metronidazole (81.2%), followed by ampicillin (75%) (Figure 1 and Figure 2). Generally, the overall resistance profiles of *C. perfringens* isolates were shocking as 74% (37 out of 50) were MDR. Additionally, three isolates, including two milk and one chicken meat, could not be treated with any of the tested antimicrobial agents. 

### 3.3. Molecular Basis of C. perfringens Resistance to Antimicrobials

A total of 85.7% (36 of 42) of the tetracycline resistant phenotypes harbored *tet* gene(s). Most tetracycline-resistant *C. perfringens* isolates harbored *tet(K)*, *tet(M)*, and *tet(L)* genes with prevalence rates of 45.2, 47.6, and 38%, respectively. Moreover, *lnu* gene(s) were found in 82.1% (23 of 28) of the lincomycin-resistant isolates. The *lnu(A)* and *lnu(B)* genes occurred in 39.2% (11/28) and 53.5% (15/28) of the lincomycin-resistant *C. perfringens* isolates, respectively. All enrofloxacin-resistant isolates (n = 27) harbored *qnr* gene(s). The *qnrA* and *qnrB* genes occurred in 74% (20/27) and 51.8% (14/27) of the enrofloxacin resistance isolates, respectively. Additionally, the *bla* gene occurred in 46.1% (12 of 26) of cefoxitin and 46.6% (7 of 15) of ampicillin resistant isolates. A total of 26 (72.2%) of the 36 erythromycin-resistant *C. perfringens* isolates carried the *erm(B)* gene (Figure 2). As expected, none of the tetracycline, lincomycin, enrofloxacin, and ampicillin or cefoxitin susceptible isolates carried *tet*, *lnu*, *qnr*, and *bla* genes, respectively. Moreover, the occurrence of relevant antibiotic resistance genes among resistant isolates was correlated with phenotypic antibiotic resistance and their molecular markers only. Meanwhile, our results confirmed that the absence of the resistance genes investigated did not predict the antimicrobial susceptibility, as some resistant strains lacked the common related resistance genes.

### 3.4. Toxinotyping and Toxin Gene Profiling of C. perfringens Isolates

All recovered *C. perfringens* isolates were confirmed as toxigenic based on dermonecrotic reactions in albino guinea pigs. Interestingly, all toxinotypes were distributed among the tested *C. perfringens* isolates in our study. The toxin gene profiling clarified that *C. perfringens* type A had an occurrence that was higher than the other types, with a percentage of 54% (27 of 50). The least prevalent types were toxinotypes C and E (8% each, 4 out of 50). Most of the toxinotypes A and C were common among the milk isolates; however, the highest prevalence of toxinotypes B was detected among the beef isolates. The toxinotypes D and E prevailed among chicken meat isolates (Figure 3). The toxin genes *cpa* (72%), *cpe* (70%), and *cpb2* (64%) were common, contrasting with the *ia* gene (8%), among the tested *C. perfringens* isolates (Figure 2). 

### 3.5. Phenotypic and Genotypic Diversity

Our isolates showed high diversity and polyclonality based on the antimicrobial resistance and toxin gene profiles. Interestingly, all the tested isolates belonged to different lineages, with the exception of two isolates: one from beef and the other from milk samples (code numbers B8D and M7D, respectively; Figure 4). 

### 3.6. Correlation Analysis between Antimicrobial Susceptibility, Toxinotypes, and Toxin Gene Profiles 

Toxinotype A was positively correlated with resistances to erythromycin, tetracycline, and the *tet(L)* gene, but it was negatively correlated with the resistance to metronidazole and the *qnrB* gene (Figure 5). We also recorded a positive correlation between toxinotype B and resistances to lincomycin, chloramephenicol, and enorfloxacin, and the *erm(B)*, *qnrA*, *qnrB,* and *bla* genes, and a negative correlation between toxinotype B and the *tet(K)* gene. Moreover, toxinotype C was positively correlated with the *qnrB* and *tet(L)* genes and negatively correlated with the resistances to tetracycline, chloramphenicol, lincomycin, and ampicillin in addition to the *bla* gene. Toxinotype D was also correlated with the resistances to imipenem and metronidazole, and to the *cpe* gene. A negative correlation was recorded between toxinotype D and the resistances to erythromycin, chloramphenicol, and cefoxitin, and the *tet(L)*, *erm(B)*, *qnrA*, and *cpb2* genes. Toxinotype E exhibited a positive correlation with the resistances to chloramphenicol and metronidazole, and the *inu(A)*, *qnrB*, and *cpb2* genes, and a negative correlation with the resistances to erythromycin and tetracycline, and the *tet(L)* gene. The *cbp2* gene was positively correlated with toxinotype E, the resistances to chloramphenicol, lincomycin, and cefoxitin, and the *inu(B), cpe**, inu(A),* and *ia* genes. The *cbp2* gene was negatively correlated with toxinotype D, the resistances to tetracycline and erythromycin, and the *etx* and *tet(M)* genes. In addition, there was a positive correlation between the *cpe* gene and toxinotype D, the resistances to tetracycline, metronidazole, cefoxitin, ampicillin, lincomycin, and chloramphenicol, and the *etx* gene, and a negative correlation between the *cpe* gene and toxinotypes E and A, the resistances to erythromycin and ampicillin, and the *cpa*, *ia*, *qnrB*, *tet(L),* and *tet(K)* genes.

## 4. Discussion

Risk-based food safety assessments are increasing worldwide. Foodborne pathogens are causing important outbreaks and diseases that have significant effects on the human health and economy. Food-poisoning outbreaks primarily comprise meat and meat products, but other food items, such as milk, may also be contaminated [45,46,47]. This crisis was compounded by the evolution of MDR/toxigenic strains [48,49,50]. Despite notable advances in the innovative next-generation therapies [51,52], the treatment failures for these pathogens were increased. *Clostridium perfringens* is an important foodborne pathogen, which causes several human and animal histotoxic and gastrointestinal diseases [53]. The high temperature used to cook meat products may inactivate the vegetative cells of *C. perfringens*; however, their spore can survive and then germinate, multiply, and produce toxins that lead to pronounced consumer health hazards [54]. On average, seven people in the United States and 50 to 100 people in the United Kingdom are killed by this foodborne illness per year [12,55]. Still, a limited number of studies have investigated this pathogen [56]. Therefore, this study attempted to break through this issue and examined the correlation between antimicrobial resistance, toxin profiles, and toxinotypes of *C. perfringens*.

The overall prevalence *C. perfringens* in our study (11.1%) was similar to that of studies in India (11%) [57] and Egypt (12.8%) [58]. In addition, our prevalence was lower than that recorded for chicken meat in China (23.1%) [59] and for American retail foods (30% and 80%) in the USA [60,61]. On the other hand, a lower prevalence was recorded in Pakistan (4%) [62] and Japan (8%) [63], and some reports did not detect *C. perfringens* in any of the examined chicken meat samples [64,65]. *C. perfringens* could be isolated from all of the samples tested (chicken meat, beef, and raw milk). The samples most infected with *C. perfringens* were chicken meat with a prevalence rate of 12.6%. A similar result was reported for Pakistan [62] and Japan [66]. Meanwhile, this contradicted a report in India [67], which reported that the highest contamination level of *C. perfringens* was recorded for goat meat. The varying prevalence of *C. perfringens* may be attributed to several factors such as differences in the hygienic conditions of the populations studied in the different studies, and the various techniques used to detect and isolate the organism in the collected samples [62]. Moreover, the variations in the hygienic practices during slaughtering, processing, and handling from production to consumption, the addition of additives, preservatives, and spices, and the stress conditions before and after the birds are slaughtered may affect the bacterial loads [68].

Several studies have referred to the increasing prevalence of MDR *C. perfringens,* which poses a serious threat to the efficient treatment for foodborne illness by limiting the therapeutic options [24,27]. The *C. perfringens* tested in this study showed frustrating susceptibility patterns, as 74% of the isolates were MDR. The antibiotics used in animal feed as growth promoters were the main causes for the evolution of *C. perfringens* resistance patterns as the bacteria become adapted as a consequence of the repeated use of antibiotics [69]. Therefore, there is an urgent need for solid guidelines defining the use of antibiotics and safe production of food products of animal origin. 

All tested resistance genes were distributed with different prevalence among the resistant strains, and none of the susceptible strains had the related resistance genes. The *tet* genes were the most common resistance genes among our tested isolates. This result parallels with the phenotypic detection of resistance as most of our isolates were resistant to tetracycline. Several previous studies have established that tetracycline resistant strains were the most common phenotypes [70,71]. The continuous use of tetracycline as a growth promoter and the presence of numerous genes associated with the resistance to tetracycline shared among different *C. perfringens* isolates may explain the high prevalence of tetracycline resistant phenotypes [72]. 

As previously documented and supporting our results, the meat and meat products are commonly infected with *C. perfringens* type A [53,73]. Although the toxinotype A prevailed (54%), our *C. perfringens* isolates belonged to different linages as shown in Figure 2 and Figure 3. The tested isolates in this manuscript, as well as in other studies, showed high heterogeneity and belonged to several toxinotypes (A, B, C, D, and E) [74,75]. The heterogeneous nature of *C. perfringens* is possibly due to their recombination, in vivo and in vitro horizontal gene transfer, evolutionary dynamism, and, to a lesser extent, host specificity. Additionally, the non-outbreak isolates and randomly selected isolates showed high diversity in contrast to the isolates from the outbreaks [60,76]. 

In this report, the *cpe* and *cpb2* genes were highly distributed among the tested isolates with prevalence rates of 70 and 64%, respectively. This was expected as all toxinotypes may potentially harbor these toxins in contrast to other toxins, which are linked to certain toxinotypes [21]. Another explanation for the high prevalence of *cpe* and *cpb2* genes was that they are carried on both plasmid and chromosome [77,78] and the plasmid encoded genes can be transferred among the isolates, especially those recovered from the food chain [79]. Notably, *C. perfringens* type A can be modified to another distinct toxinotype upon acquisition of plasmid encoding toxins [80,81]. Therefore, the possibility of acquisition of a single toxin, such as CPE, is limited, supporting our results, once we recorded a negative correlation between the *cpe* gene and toxinotypes A and E. Several authors have correlated antimicrobial resistance patterns to only the administered antibiotics in animal feeds [82,83]. Moreover, the correlation between virulence and resistance was assessed in several other studies [84,85,86], and the genetic diversity of *C. perfringens* complicates these correlations [87]. In our report, we found weak positive and negative correlations. However, these correlations were variable, and we could not reach solid conclusions or fixed links. The treatment with antibiotics may increase the virulence expression such as of the *cpb2* gene, due to antibiotic-induced ribosomal frameshifting [83]. Therefore, the antimicrobial resistance patterns may be unreliable markers to be correlated with the toxin profiles or toxinotypes of *C. perfringens*.

## 5. Conclusions

Beef, chicken meat, and raw milk samples from Egyptian supermarkets could be potentially contaminated with *C. perfringens*, especially with the toxinotype A, and the examined samples exceeded the permissible limits for *C. perfringens,* reflecting poor storage or poor processing conditions. Moreover, our results offer further evidence on the emergence of MDR *C. perfringens* strains. The antimicrobial resistance and toxin gene profiles expand knowledge on the high diversity and polyclonality of *C. perfringens* isolates. It is therefore recommended that the food safety standards and frequent inspections of the sanitary measures in supermarkets should be adequately enforced for efficient prevention of *C. perfringens* foodborne-associated infections among humans and animals. Moreover, control measures for proactive antimicrobial agents should be defined to limit the spread of MDR strains. The small number of the isolated strains (50) and the lack of whole-genome sequencing in this study prevent us from putting up solid correlations between antimicrobial resistance and toxinotypes of *C. perfringens*. Therefore, further studies in this point must be continued to provide a strong plane for the infection control protocols for *C. perfringens*. 

## Figures and Tables

**Figure 1 biology-11-00551-f001:**
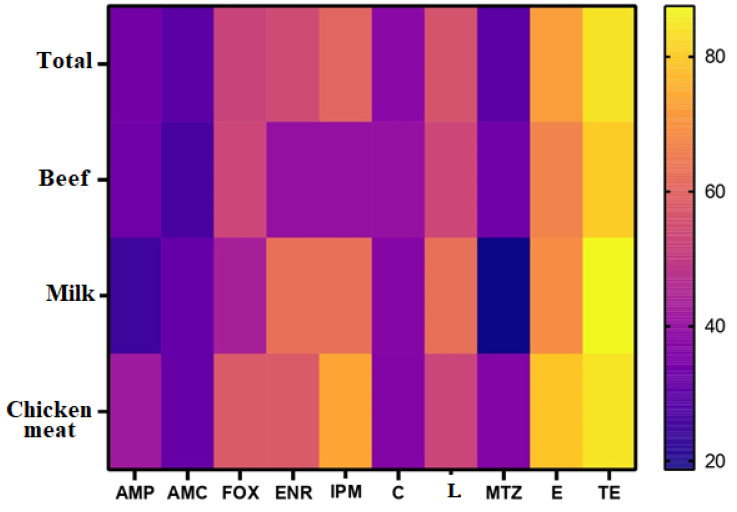
Frequency of resistance of *C. perfringens* isolates from beef, milk, and chicken meat samples to antimicrobials. AMP, ampicillin; AMC, amoxicillin/clavulanic acid; FOX, cefoxitin; ENR, enrofloxacin; IPM, imipenem; C, chloramphenicol; L, lincomycin; MTZ, metronidazole; E, erythromycin; TE, tetracycline. The percentages of resistance to antimicrobials are color-coded on the right of the figure.

**Figure 2 biology-11-00551-f002:**
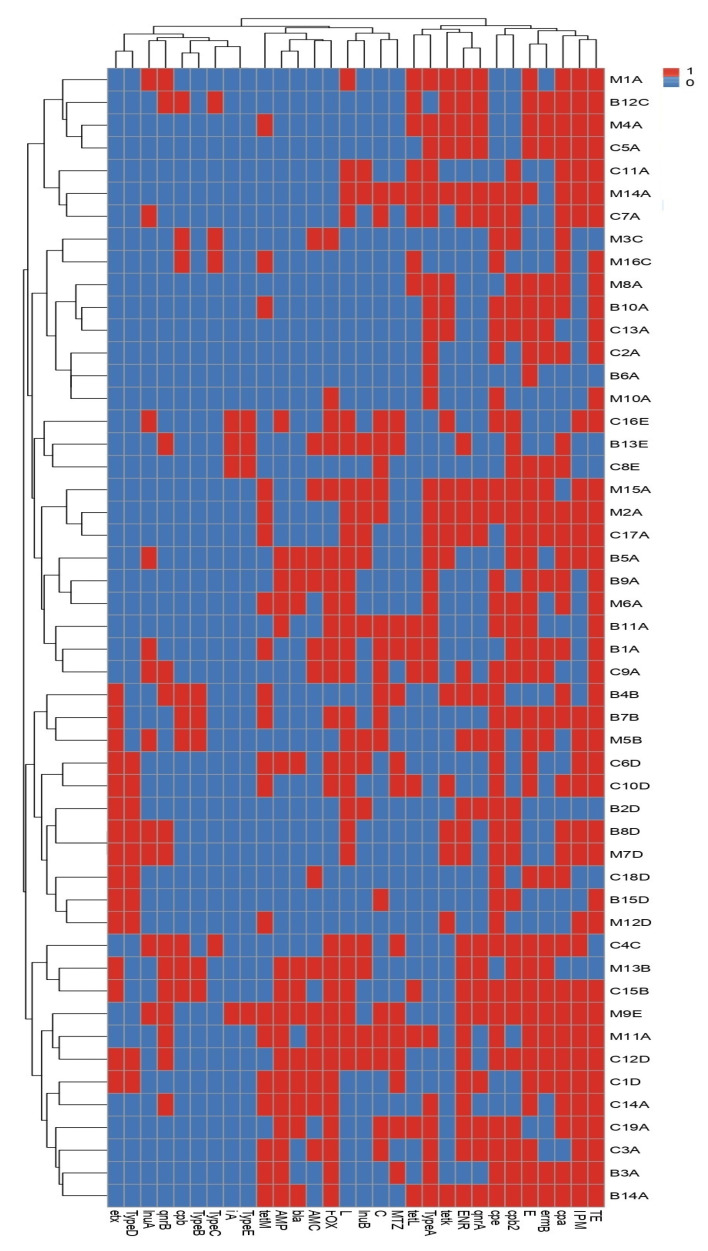
Heat map and hierarchical clustering of the examined *C. perfringens* isolates based on the occurrence of antimicrobial resistance, antibiotic resistance, and toxin genes and toxinotypes. In the heat map, red and blue colors refer to the resistance/sensitivity to an antimicrobial agent and to the presence/absence of an antibiotic resistance, the toxin gene and toxinotype, respectively. The code numbers on the right of the heat map refer to the isolate numbers for beef (B), chicken meat (C), and milk (M) samples. AMP, ampicillin; AMC, amoxicillin/clavulanic acid; FOX, cefoxitin; ENR, enrofloxacin; IPM, imipenem; C, chloramphenicol; L, lincomycin; MTZ, metronidazole; E, erythromycin; TE, tetracycline. The *tet(K)*, *tet(L)*, and *tet(M)*; *lnu(A)* and *lnu(B)*; *erm(B)*; *bla*; and *qnrA* and *qnrB* are genes associated with tetracycline, lincomycin, erythromycin, β-lactams, and enrofloxacin resistances, respectively. The *cpa, cpb, etx, ia,* and *cpe* are *C. perfringens* alpha, beta, epsilon, iota, and enterotoxin genes, respectively, and *cpb2* is the *C. perfringens* beta2 toxin gene.

**Figure 3 biology-11-00551-f003:**
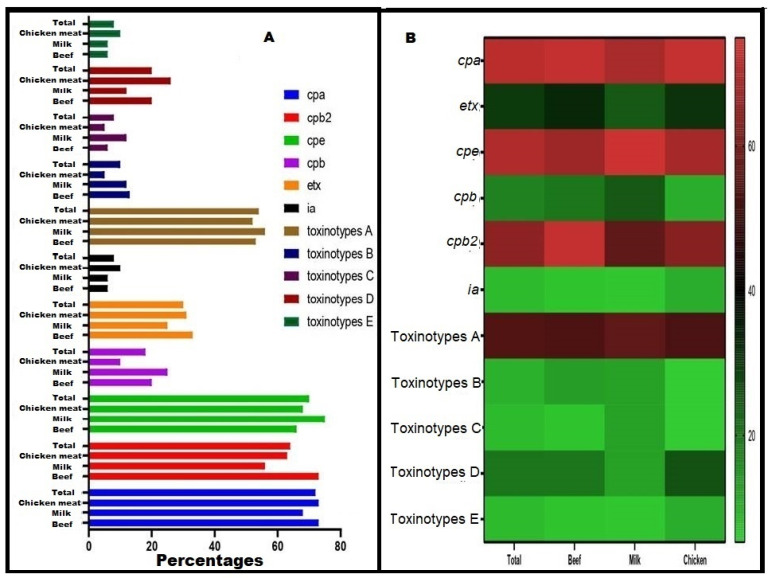
Distribution of toxinotypes and toxin genes among *C. perfringens* from chicken meat, milk, and beef samples. (**A**) Columns style using Graphpad prism, which showes the percentages of *C. perfringens* toxins and toxinotypes from each sample type. (**B**) Heat map style, in which the percentages of toxinotypes and toxin genes are color-coded on the right of the figure. The *cpa, cpb, etx, ia,* and *cpe* are *C. perfringens* alpha, beta, epsilon, iota, and enterotoxin genes, respectively; *cpb2* is the *C. perfringens* beta2 toxin gene.

**Figure 4 biology-11-00551-f004:**
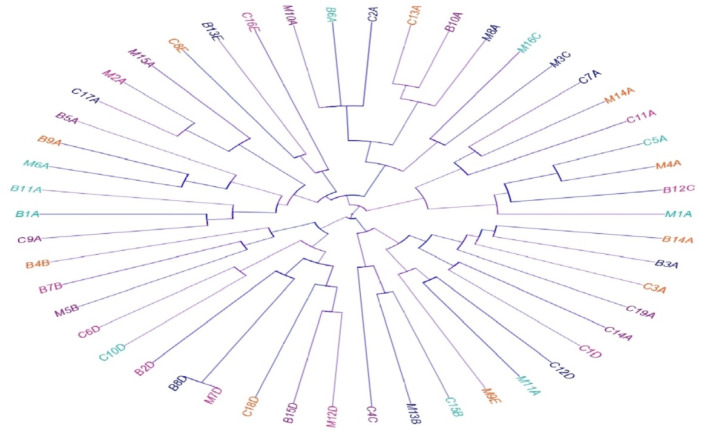
Dendrogram showing the relatedness of *C. perfringens* isolated from beef (B), chicken meat (C), and milk (M) samples as determined by the antimicrobial resistance and toxin gene profiles. The *C. perfringens* toxinotypes are indicated with different colors in the dendrogram to denote the specificity of various toxinotypes.

**Figure 5 biology-11-00551-f005:**
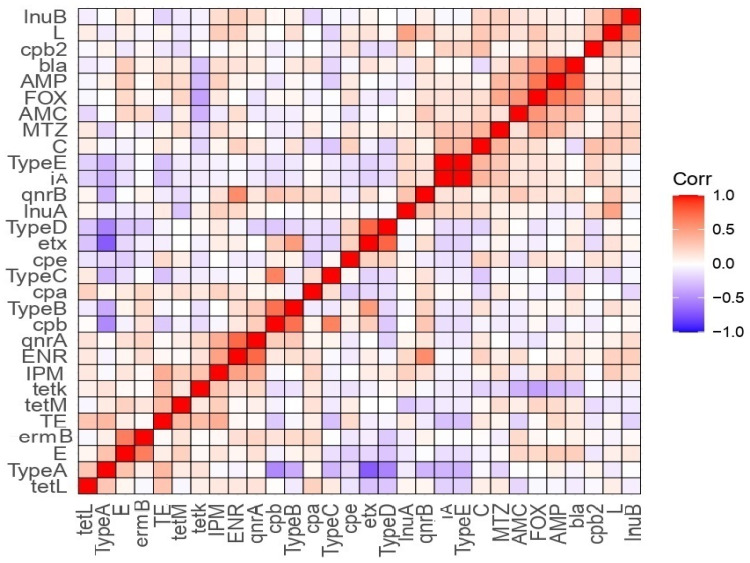
Correlation (*r*) between antimicrobial resistance, antibiotic resistance, and toxin genes and toxinotypes of *C. perfringens* isolates from different sample types. Red and blue colors indicate positive and negative correlations, respectively. The color key refers to correlation coefficient (*r*). The darker red and blue colors imply stronger positive (R = 0.5:1) and negative (R = −0.5:−1) correlations, respectively. AMP, ampicillin; AMC, amoxicillin/clavulanic acid; FOX, cefoxitin; ENR, enrofloxacin; IPM, imipenem; C, chloramphenicol; L, lincomycin; MTZ, metronidazole; E, erythromycin; TE, tetracycline. The *tet(K)*, *tet(L)*, and *tet(M)*; *lnu(A)* and *lnu(B)*; *erm(B)*; *bla*; and *qnrA* and *qnrB* are genes associated with tetracycline, lincomycin, erythromycin, β-lactams, and enrofloxacin resistances, respectively. The *cpa, cpb, etx, ia,* and *cpe* are *C. perfringens* alpha, beta, epsilon, iota, and enterotoxin genes, respectively, and *cpb2* is the *C. perfringens* beta2 toxin gene.

**Table 1 biology-11-00551-t001:** Targeted resistance and toxin genes of *C. perfringens* and their primer sequences, expected amplicon sizes, and annealing temperatures.

Target Gene	Primer Sequence (5′-3′)	Amplicon Size (bp)	Annealing Temperature (°C)	Reference
*tet(K)*	F: TTATGGTGGTTGTAGCTAGAAA R: AAAGGGTTAGAAACTCTTGAAA	382	50	[40]
*tet(L)*	F: ATAAATTGTTTCGGGTCGGTAAT R: AACCAGCCAACTAATGACAATGAT	1077	50	[38]
*tet(M)*	F: ACAGAAAGCTTATTATATAAC R: TGGCGTGTCTATGATGTTCAC	171	55	[39]
*lnu(A)*	F: GGTGGCTGGGGGGTAGATGTATTAACTGG R: GCTTCTTTTGAAATACATGGTATTTTTCGATC	323	54	[39]
*lnu(B)*	F: CCTACCTATTGTTTGTGGAA R: ATAACGTTACTCTCCTATTC	906	45	[39]
*erm(B)*	F: GAAAAGGTACTCAACCAAATA R: AGTAACGGTACTTAAATTGTTTAC	638	57	[28]
*bla*	F: ATGAAAGAAGTTCAAAAATATTTAGAG R: TTAGTGCCAATTGTTCATGATGG	780	50	[42]
*qnrA*	F: AGAGGATTTCTCACGCCAGG R: TGCCAGGCACAGATCTTGAC	580	54	[41]
*qnrB*	F: GGMATHGAAATTCGCCACTG R: TTTGCYGYYCGCCAGTCGAA	264	54	[41]
*cpa*	F: GCTAATGTTACTGCCGTTGA R: CCTCTGATACATCGTGTAAG	324	54	[41]
*cpb*	F: GCGAATATGCTGAATCATCTA R: GCAGGAACATTAGTATATCTTC	196	54	[43]
*etx*	F: GCGGTGATATCCATCTATTC R: CCACTTACTTGTCCTACTAAC	655	54	[43]
*iA*	F: ACTACTCTCAGACAAGACAG R: CTTTCCTTCTATTACTATACG	446	54	[43]
*cpe*	F: GGAGATGGTTGGATATTAGG R: GGACCAGCAGTTGTAGATA	233	54	[43]
*cpb2*	F: AGATTTTAAATATGATCCTAACC R: CAATACCCTTCACCAAATACTC	567	54	[44]

## Data Availability

All data generated or analyzed during this study are included in the published article and the Appendix A.

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
