# Peer review of "Clostridium perfringens Associated with Foodborne Infections of Animal Origins: Insights into Prevalence, Antimicrobial Resistance, Toxin Genes Profiles, and Toxinotypes"

_biology, 2022, doi:10.3390/biology11040551_

Round 1

Reviewer 1 Report

1- The authors described in the materials and methods section the phenotypic characterization of the isolated C. perfringens strains including colonies description, I would recommend including a plate image showing clearly the described colonies

2- The authors again mentioned that they performed sequencing of 16S rRNA genes so they should include a pairwise alignment of one of the sequenced genes at least to confirm it belongs to C. perfringens and they must deposit the sequencing files they got in the supplementary files

3- The authors must find a professional way to show figures 1,2 and 4 like graphpad-prism

4- In Figure 3, the authors showed a heat map with a side color scale, while the heat map is only 2 colors (red and blue), I am wondering about the information we can extract from the heat map and whether it is the suitable way to show the result

5- Regarding the correlation analysis between antimicrobial susceptibility, toxinotypes and toxin gene profiles. In the abstract, the authors mentioned that they could not conclude the type of correlation but then in the results, they described some kind of variable correlations, how significant are these correlations and they must mention some references for studies that support this conclusion.

Author Response

First of all, many thanks for the reviewer comments and the opportunity to further revise the paper. We would like to thank the reviewer for his raised and thorough comments and we go precisely through the comments of the reviewer to meet his expectations.

1- The authors described in the materials and methods section the phenotypic characterization of the isolated C. perfringens strains including colonies description, I would recommend including a plate image showing clearly the described colonies.

Response to reviewer: In accordance with the reviewer point of view, a plate image of the described colonies of C. perfringens is introduced as supplementary Figure 1.

2- The authors again mentioned that they performed sequencing of 16S rRNA genes so they should include a pairwise alignment of one of the sequenced genes at least to confirm it belongs to C. perfringens and they must deposit the sequencing files they got in the supplementary files

Response to reviewer: Thank you for this comment; however, there is a misunderstanding in this point as we used species-specific PCR primers (ClPER) targeting a unique region of C. perfringens 16S rRNA gene. The genetic identification of C. perfringens was confirmed by PCR amplification not the sequencing of this region according to the following reference:

Kikuchi, E.; Miyamoto, Y.; Narushima, S.; Itoh, K. Design of species-specific primers to identify 13 species of Clostridium harbored in human intestinal tracts. Microbiol Immunol. 2002, 46, 353– 358. Additionally, we rephrased this sentence in the material and methods section to be more clearer.

3- The authors must find a professional way to show Figures 1, 2 and 4 like graphpad-prism

Response to reviewer: Thank you for this comment. In accordance with your recommendation, Figures 2 and 4 were modified using the graphpad-prim and depending on the recommendations of  other reviewers, Figure 1 was deleted and more information concerning the data in this Figure were inserted in the revised manuscript.

4- In Figure 3, the authors showed a heat map with a side color scale, while the heat map is only 2 colors (red and blue), I am wondering about the information we can extract from the heat map and whether it is the suitable way to show the result.

Response to reviewer: Thank you for this comment. We totally agreed with the reviewer comment; therefore, we changed the colors code and we added more information in the text parallel with this dendrogram to explain the results in a good way. 

5- Regarding the correlation analysis between antimicrobial susceptibility, toxinotypes and toxin gene profiles. In the abstract, the authors mentioned that they could not conclude the type of correlation but then in the results, they described some kind of variable correlations, how significant are these correlations and they must mention some references for studies that support this conclusion.

Response to reviewer: Thank you for this comment. We found certain correlations between antimicrobial susceptibility, toxinotypes and toxin gene profiles; meanwhile, these correlations were weak and variable. We mentioned and discussed these correlation in the results and discussion sections, but the overall correlation couldn’t provide us with solid results. Therefore, we couldn’t conclude fixed correlations. Additionally, we rephrased these information in the abstract and more information were compared with several new studies at the end of discussion section.

Reviewer 2 Report

This study found increased prevalence of CP obtained from various sources and higher multi-drug resistant patterns. Among many foodborne pathogens, CP is neglected, and authors have found great findings on this aspect. It is a great piece of evidence for researchers who will explore on controlling this pathogen. Not only that, but this study also isolated among 450 samples which not many studies consider testing. Therefore, it is solid evidence for this pathogen which is a greatest concern globally. I highly consider authors to provide “out of 450 samples- 50 were contaminated with CP… in the abstract”. Also, provide previous prevalence studies on this pathogen.
Please change the wordings such as 'tried'. Instead, explored can be a better one. 
Italicize genes in L41-42 and rest of the manuscript. Also, provide more information on what these genes function in introduction as well.
Italicize all bacterial names (:54 Salmonella) and please spell out positive.
Introduction is well justified.
Also, the bar figures must be changed from 3D to normal as they don’t sound scientific. 
Line 196: remove as shown and put figures in parenthesis
All the results and discussions are well written and justified. It is a very nice research study.

Author Response

This study found increased prevalence of CP obtained from various sources and higher multi-drug resistant patterns. Among many foodborne pathogens, CP is neglected, and authors have found great findings on this aspect. It is a great piece of evidence for researchers who will explore on controlling this pathogen. Not only that, but this study also isolated among 450 samples which not many studies consider testing. Therefore, it is solid evidence for this pathogen which is a greatest concern globally.

First of all, many thanks for the reviewer comments and the opportunity to further revise the paper. We would like to thank the reviewer for his raised and thorough comments and we go precisely through the comments of the reviewer to meet his expectations.

1- I highly consider authors to provide “out of 450 samples- 50 were contaminated with CP… in the abstract”. Also, provide previous prevalence studies on this pathogen.

Response to reviewer: Thank you for this valuable comment. In accordance with this comment, we found that it is essential to add this information in the abstract and it was already added. Moreover, additional previous studies on the prevalence of this pathogen were compared with our results in the second paragraph of the discussion section.

 2- Please change the wordings such as 'tried'. Instead, explored can be a better one. 

Response to reviewer: Thank you for this comment. It was modified as the reviewer asked.

 3- Italicize genes in L41-42 and rest of the manuscript. Also, provide more information on what these genes function in introduction as well.

Response to reviewer: we apologized for these mistakes. All these genes were italicized and we provided the requested information as well.

 4- Italicize all bacterial names (:54 Salmonella) and please spell out positive.
Introduction is well justified.

Response to reviewer: Thanks for these corrections. We did the corrections as requested.

 5- Also, the bar figures must be changed from 3D to normal as they don’t sound scientific. 

Response to reviewer: In accordance with your recommendations as well as other reviewers' point of views, these figure were changed using the graphpad-prism without 3D design.

6- Line 196: remove as shown and put figures in parenthesis.

Response to reviewer: Thanks for this comment. We rephrase this sentence in accordance with the reviewer recommendation.

 7- All the results and discussions are well written and justified. It is a very nice research study.

Response to reviewer : Thanks  a lot for these valuable positive comments.

Reviewer 3 Report

This manuscript has screened Clostridium perfringens foodborne associated infections in Egypt (including prevalence, antimicrobial resistance, toxin genes 3 profiles and toxinotypes). The overall content of manuscript is acceptable. I would like to recommend this manuscript after incorporating revisions. The following aspects should be addressed:

  1. Title: I think the origin of samples should be added, i.e.
  2. Section 2.1: standard unit should be used, e.g. Line 100 15 minutes should be changed to 15 min
  3. Section 2.3 The C. perfringens ATCC 128 3626 and E. coli ATCC 25922 strains were used as positive and negative controls, respec-129 tively during all PCR runs. Please explain why E. coli ATCC 25922 strain was chosen as negative control.
  4. Figure 1: the current bars in Fig. 1 are not well presented. For example, number of samples has been reported in Materials and Methods. It is no longer necessary to present in the figure. Secondly, the prevalence rate can be easily derived from number of isolates. I suggest removing this figure. The current data can be easily reported in the main body of manuscript.
  5. Figure 2: error bar should be added.

Author Response

This manuscript has screened Clostridium perfringens foodborne associated infections in Egypt (including prevalence, antimicrobial resistance, toxin genes 3 profiles and toxinotypes). The overall content of manuscript is acceptable. I would like to recommend this manuscript after incorporating revisions. The following aspects should be addressed:

First of all, many thanks for the reviewer comments and the opportunity to further revise the paper. We would like to thank the reviewer for his raised and thorough comments and we go precisely through the comments of the reviewer to meet his expectations.

1- Title: I think the origin of samples should be added, i.e.

Response to reviewer: Thanks a lot, for this comment. We totally agreed with the reviewer comment and the origin of the samples was added in the title.

 2- Section 2.1: standard unit should be used, e.g. Line 100 15 minutes should be changed to 15 min.

Response to reviewer: We apologized for this mistake and it was corrected.

 3- Section 2.3 The C. perfringens ATCC 128 3626 and E. coli ATCC 25922 strains were used as positive and negative controls, respectively during all PCR runs. Please explain why E. coli ATCC 25922 strain was chosen as negative control.

Response to reviewer: Thank you for your comment. The validation of our PCR results relied on C. perfringens ATCC 3626 and E. coli ATCC 25922 strains as positive and negative controls, respectively for ClPER gene in their respective PCR assays and we repeated all PCR runs three times to confirm our results. The gel electrophoresis of PCR products confirmed the amplification of the target genes` sequences for all positive controls. Bands of the expected sizes corresponding to the gene fragments were observed for the positive controls, but no bands were observed for the negative controls. The positive controls were provided from the National Laboratory for Veterinary Quality Control on Poultry Production (NLQP), Animal Health Research Institute, Giza, Egypt. On the other hand, we selected the DNA of standard strain of different species “E. coli ATCC 25922” other than C. perferingens, which was available in our laboratory as a negative control.

Regarding the toxin and resistance genes, all PCR runs were performed with relevant PCR positive controls (DNAs from C. perfringens isolates previously confirmed to harbor sequences for any of the target genes by gel electrophoresis of PCR products). These positive controls were provided from the National Laboratory for Veterinary Quality Control on Poultry Production (NLQP). Additionally, sterile saline was used as a negative control. This was added in the revised manuscript.

4- Figure 1: the current bars in Fig. 1 are not well presented. For example, number of samples has been reported in Materials and Methods. It is no longer necessary to present in the figure. Secondly, the prevalence rate can be easily derived from number of isolates. I suggest removing this figure. The current data can be easily reported in the main body of manuscript.

Response to reviewer: Thank you for this positive criticism and we totally agreed with the reviewer point of view; therefore, we removed Figure 1 and added more information in the text, which present the data in this Figure.

 5- Figure 2: Error bar should be added.

Response to reviewer: According to this comment and other reviewer comments, Figure 1 was removed and Figures 2 and 4 were modified using the graphpad-prism program.

Reviewer 4 Report

In this manuscript, the authors present the detection of C. perfringens from food samples and the characterization of the isolates (resistance pattern, resistance genes, toxin genes and toxinotypes.

I would suggest the authors to have their manuscript proof read by an English professional to improve its quality.

L41-42: Genes have to be in italics.

L43: Please define the genes.

L84: Please put lnu in italics.

L93-94: Please provide more information about the samples origin (number supermarkets in both governorates, number of each kind of samples in each supermarket, etc.).

L94: Shouldn’t it be Port Said?

L97: far->for ?

L151-166: Please precise the strain that was used as positive control for the PCR detection of toxins and resistance genes.

L245: If I understand correctly the figure 4, toxinotype D and E  are prominent among chicken meat isolates not the milk ones. Can you double check, please?

L245: Toxinotypes are only shown in figure 4. Please remove the mention of figure 3.

L247: please add a mention to figure 3.

L254 and 262: susceptibility-> resistance.

L318: Please precise in which countries these data come from.

Figure 2: Please provide the resistance threshold for each antimicrobial that was used for the characterization of the strains.

Author Response

In this manuscript, the authors present the detection of C. perfringens from food samples and the characterization of the isolates (resistance pattern, resistance genes, toxin genes and toxinotypes.

First of all, many thanks for the reviewer comments and the opportunity to further revise the paper. We would like to thank the reviewer for his raised and thorough comments and we go precisely through the comments of the reviewer to meet his expectations.

I would suggest the authors to have their manuscript proof read by an English professional to improve its quality.

Response to reviewer: Thank you for your recommendation. We already sent our manuscript to an international English proof editing center to improve its quality and the certificate for this proof editing is uploaded with the response to the reviewer.

 L41-42: Genes have to be in italics.

Response to reviewer: We apologized for this mistake and it was corrected.

L43: Please define the genes.

Response to reviewer: They were defined.

 L84: Please put lnu in italics.

Response to reviewer: We apologized for this mistake and it was corrected.

 L93-94: Please provide more information about the samples origin (number supermarkets in both governorates, number of each kind of samples in each supermarket, etc.).

Response to reviewer: Thank you for this positive criticism and we totally agreed with the reviewer point of view; therefore, we added more information in the material and methods section about the sample origins.

 L94: Shouldn’t it be Port Said?

Response to reviewer: Thank you for this comment. It was corrected.

 L97: far->for ?

Response to reviewer: We apologized for this mistake and it was corrected.

 L151-166: Please precise the strain that was used as positive control for the PCR detection of toxins and resistance genes.

Response to reviewer: Thank you for your comment. The validation of our PCR results relied on C. perfringens ATCC 3626 and E. coli ATCC 25922 strains as positive and negative controls, respectively for ClPER gene in their respective PCR assays and we repeated all PCR runs three times to confirm our results. The gel electrophoresis of PCR products confirmed the amplification of the target genes` sequences for all positive controls. Bands of the expected sizes corresponding to the gene fragments were observed for the positive controls, but no bands were observed for the negative controls. The positive controls were provided from the National Laboratory for Veterinary Quality Control on Poultry Production (NLQP), Animal Health Research Institute, Giza, Egypt. Regarding the toxin and resistance genes, all PCR runs were performed with relevant PCR positive controls (DNAs from C. perfringens isolates previously confirmed to harbor sequences for any of the target genes by gel electrophoresis of PCR products). These positive controls were provided from the National Laboratory for Veterinary Quality Control on Poultry Production (NLQP). Additionally, sterile saline was used as a negative control. This was added in the revised manuscript.

 L245: If I understand correctly the figure 4, toxinotype D and E are prominent among chicken meat isolates not the milk ones. Can you double check, please?.

Response to reviewer: Thank you for your valuable comment and good reviewing. In accordance with your good vision and our double check, we found that toxinotype D and E  are prominent among chicken meat isolates and therefore, it was corrected in the revised manuscript.

 L245: Toxinotypes are only shown in figure 4. Please remove the mention of Figure 3.

Response to reviewer: Thank you for your good reviewing. The mention of Figure 3 was removed.

 L247: please add a mention to Figure 3.

Response to reviewer:  Thank you for your comment. The mention was added.

 L254 and 262: susceptibility-> resistance.

Response to reviewer:  Thank you for your comment. It was rephrased according to the reviewer advice.

 L318: Please precise in which countries these data come from.

Response to reviewer:  Thank you for your comment. It was added.

Figure 2: Please provide the resistance threshold for each antimicrobial that was used for the characterization of the strains.

Response to reviewer: Thank you for your comment. The resistance threshold for all the tested antimicrobial agents was provided in a simple supplementary Table (Table S1) in the revised manuscript.

Round 2

Reviewer 1 Report

Thanks to authors for addressing most of the points, I have mentioned.

I have some minor points

I would like to ask them to change Figure 3. Distribution of toxinotypes and toxin genes among C. perfringens from heat map style to columns style using Graphpad prism to add information about meat source , also in the figure caption change it to figure 2 not figure 3

Author Response

I would like to ask them to change Figure 3. Distribution of toxinotypes and toxin genes among C. perfringens from heat map style to columns style using Graphpad prism to add information about meat source , also in the figure caption change it to figure 2 not figure 3

Response:  thank you for the good reviewing and doing of your best effort to allow us to explore the results of our manuscript in suitable form. We totally agree with your point of view and in accordance,  we added column style figure in addition to the heat map style as figure 3A and 3B. Additionally, we cannot move this figure to be figure 2 as the antimicrobial resistances were discussed before the toxinotypes in all parts of our manuscript